# Improvement of Low Luminance Visual Acuity in Patients with Chronic Central Serous Chorioretinopathy after Half-Dose Verteporfin Photodynamic Therapy

**DOI:** 10.3390/jcm9123980

**Published:** 2020-12-09

**Authors:** Kyoko Fujita, Kei Shinoda, Yutaka Imamura, Celso Soiti Matsumoto, Koichi Oda

**Affiliations:** 1Department of Opthalmology, Aichi Medical University School of Medicine, Aichi 480-1195, Japan; 2Department of Ophthalmology, Saitama Medical University School of Medicine, Saitama 350-0495, Japan; shinodak@med.teikyo-u.ac.jp; 3Department of Ophthalmology, Teikyo University School of Medicine, University Hospital Mizonokuchi, Kanagawa 213-8507, Japan; yimamura.ny@gmail.com; 4Department of Ophthalmology, Teikyo University School of Medicine, University Hospital Itabashi, Tokyo 173-8606, Japan; soiti@icloud.com; 5Department of Communication, Tokyo Woman’s Christian University, Tokyo 167-8585, Japan; k-oda@lab.twcu.ac.jp

**Keywords:** central serous chorioretinopathy, low luminance visual acuity, photodynamic therapy, ellipsoid zone

## Abstract

Patients with central serous chorioretinopathy (CSC) often complain of visual difficulties under low luminance conditions. In this study, we evaluated low luminance visual acuity (LLVA) after half-dose verteporfin photodynamic therapy (hdPDT) in eyes with CSC. The study included eight eyes of eight patients with chronic CSC that underwent hdPDT. The best-corrected visual acuity, LLVA and optical coherence tomography (OCT) findings were evaluated at baseline, and at 1, 3, 6, 9, and 12 months after hdPDT. LLVA was measured at six levels of background luminance. Serous retinal detachment was completely resolved in all eyes. Although the mean LLVA at 1 month did not improve significantly compared to baseline at all luminance levels, significant improvements were observed at 3, 6, 9, and 12 months (*p* < 0.05). In OCT images, although the ellipsoid zone was not detectable in all eyes before hdPDT, it was observed in 2 eyes at 1 month, in 7 eyes at 3 months, and in all eyes from 6 months. The LLVA and the ellipsoid zone improved similarly with postoperative time courses. In conclusion, hdPDT improves LLVA in eyes with chronic CSC. The recovery of the ellipsoid zone may play a part in improving LLVA after hdPDT.

## 1. Introduction

Several factors influence visual acuity (VA), such as pupil size, object distance, target contrast, and the test object [1]. Several studies have reported a sigmoid relationship between VA and logarithm of illumination [1,2]. At low luminance, VA increases very slowly with an increase in logarithm of illumination, and this is associated with rod-dominated function. At high luminance, VA increases rapidly with an increase in the logarithm of illumination until plateau is reached, and this is a cone-dominant function. Therefore, VA may vary enormously under different luminance conditions in daily life, and the conventional VA test reflects only a partial aspect of VA [3,4].

Central serous chorioretinopathy (CSC) is characterized by serous retinal detachment (SRD) and retinal pigment epithelial detachment at the macula. The SRD often resolves spontaneously but sometimes recurs or becomes chronic [5]. Although VA is preserved relatively well in spite of SRD in the macula, patients often complain of various visual disturbances such as metamorphopsia, relative central scotoma, and micropsia [5], which may impact patients’ quality of life [6]. In addition to these symptoms, patients with CSC often complain of difficulty to see under low luminance conditions such as dawn, evening, and rainy days. Such visual disturbances are usually not detected using conventional VA tests. In a previous report, we measured low luminance visual acuity (LLVA) of CSC patients with visual disturbance using test charts with low background luminance [7]. The results showed that in CSC eyes, LLVA worsened with a decrease in background luminance. These results indicated that CSC patients had visual difficulties in an environment with lower luminance.

Half-dose verteporfin photodynamic therapy (hdPDT) proved to be an effective treatment option for chronic CSC, not only for achieving anatomical success but also for improving visual functions such as visual acuity, retinal sensitivity, and metamorphopsia [8,9,10,11]. However, there is no previous report that evaluates LLVA after hdPDT. The purpose of this study was to determine the LLVA of patients with chronic CSC after undergoing hdPDT.

## 2. Subjects and Methods

Eight eyes of eight patients diagnosed with chronic CSC, defined as showing symptoms for at least 6 months, who underwent hdPDT, were studied. Inclusion criteria included the presence of SRD involving the fovea depicted on spectral-domain optical coherence tomography (OCT) (SD-OCT; Heidelberg Engineering, Heidelberg, Germany) and abnormally dilated choroidal vasculature with hyperpermeability observed on indocyanine green angiography (ICGA) (TRC-50IX; Topcon Corp., Tokyo, Japan) before hdPDT. Patients who had other retinal diseases, including age-related macular degeneration, polypoidal choroidal vasculopathy, pathologic myopia, and tilted disc syndrome, were excluded. Patients who had undergone focal thermal laser photocoagulation or intravitreal injection of vascular endothelial growth factor inhibitors for CSC within 3 months before PDT were also excluded.

Clinical examination was performed before and at 1, 3, 6, 9, and 12 months after treatment with hdPDT. Examination included the conventional best-corrected visual acuity (BCVA) test, slit-lamp biomicroscopy, indirect funduscopy, OCT, and LLVA tests at all visits. The conventional BCVA was measured with a Landolt C chart using standard retro-illumination. The methods of the LLVA tests have been reported previously [7]. Briefly, the LLVA charts were displayed on a monitor (Sony GDMF500). The Landolt C rings used in the test followed the design rule of the Early Treatment Diabetic Retinopathy Study charts. Six levels of background luminance were tested, namely, 78.20, 31.87, 11.37, 4.14, 1.30, and 0.37 cd/m^2^. The luminance of the Landolt C rings was kept as close to zero cd/m^2^ as possible, and the contrast in all conditions approached 100 percent. Identification of the largest Landolt C ring at a distance of 308 cm represented VA of 0.7 logarithm of the minimum angle of resolution (logMAR). The ring size was reduced in steps of 0.1 logMAR. LLVA was measured starting from the lowest background and increasing sequentially to higher background luminance.

Fluorescein angiography and ICGA were performed before hdPDT. In the current study, the retinal structures at the fovea were evaluated using the horizontal B-scan cross-sectional images of SD-OCT. Restoration of the photoreceptor layer was defined as reconstruction of the continuous back-reflection lines corresponding to the ellipsoid zone and the interdigitation zone at the central fovea. Qualitative OCT findings were evaluated independently by two retina specialists (KF and KS). When there were discrepancies between the two retina specialists, a third specialist (YI) made the final decision.

PDT was performed using half-dose verteporfin (Visudyne; Novartis AG, Bulach, Switzerland). Briefly, 3 mg/m^2^ of verteporfin was infused over 10 min. At 15 min after beginning of the infusion, laser treatment was started. The total light energy delivered to the area of hyperpermeability detected on the ICGA images was 50 J/cm^2^. The spot size covered the area of dilated and congested choroidal vessels shown on the ICGA images.

For statistical analysis, conventional decimal BCVA was converted to logMAR. Wilcoxon signed-rank sum test was used to determine the significance of improvement in function after the hdPDT. A *p* value less than 0.05 was considered statistically significant.

All subjects gave their informed consent for inclusion before they participated in the study. The study was conducted in accordance with the Declaration of Helsinki, and the protocol was approved by the Ethics Committee of Nihon University Hospital (No. 20160308).

## 3. Results

All eight CSC patients studied were men, with mean (± standard deviation) age of 43.8 ± 6.74 (range 33 to 56) years. The conventional BCVA improved from −0.06 ± 0.10 logMAR at baseline to −0.16 ± 0.03 logMAR at 1 year (*p* = 0.019). The results of OCT findings are presented in Table 1.

The SRD was completely resolved at 1 month in 6 eyes, and at 3 months in 2 eyes (Table 1). 

The changes in LLVA at 6 different levels of background luminance during the follow-up period of each patient are shown in Table 2 and Figure 1. Although the LLVA at 1 month did not improve significantly compared with baseline at all the luminance levels, significant improvements were observed at 3, 6, 9, and 12 months.

Ellipsoid zone and interdigitation zone were not detectable in all the eyes before hdPDT (Table 1 and Figure 1). Ellipsoid zone at the fovea was observed in 2 of 8 eyes at 1 month, in 7 of 8 eyes at 3 months, and in all 8 eyes from 6 months. The interdigitation zone was first detected at 9 months in 2 eyes, and continuous interdigitation zone was seen at 12 months in 6 eyes (Figure 2).

None of the patients developed any systemic or ocular adverse event, and there was no severe vision loss associated with the treatment.

## 4. Discussion

The mean LLVA was measured at six different levels of background luminance after hdPDT in this study. Although the LLVA did not improve from baseline at 1 month for all the luminance levels, significantly improvements were observed at 3, 6, 9, and 12 months. The improvement of LLVA appeared to be parallel to the recovery of the ellipsoid zone observed on OCT.

In our previous study of patients with CSC, we found that LLVA worsened with decrease in background luminance [7]. This result suggests that patients with CSC have more visual difficulties in environment of low luminance. These visual disturbances may affect the patients’ daily activities such as driving at dawn or at night and reading under dim light. The current results showed that hdPDT for chronic CSC was effective in improving patients’ visual difficulties in low luminance environment.

Numerous studies have shown favorable results of hdPDT for CSC, both anatomically and functionally [8,9,10,11]. To date, no studies have evaluated LLVA after hdPDT. In this study, we demonstrated improvement of LLVA after hdPDT.

Several reports demonstrated [12,13,14] that LLVA was useful to detect visual deterioration in early and intermediate age-related macular degeneration. Sunness et al. [12]. reported that LLVA was reduced significantly in eyes with non-foveal geographic atrophy despite good BCVA, and that LLVA predicted the risk of future VA loss. Since most CSC patients show good VA when assessed by conventional tests, it is difficult for clinicians to decide the optimal timing of PDT, which is known to effectively improve conventional VA. The LLVA deficiency may reflect patients’ complaints of visual difficulties unexplainable by conventional VA test. Therefore, LLVA would assist clinicians in deciding the timing of treatment.

Several studies have shown that the status of the ellipsoid zone is associated with VA in CSC patients [10,15,16]. Piccolino et al. [15] showed that impairment of the foveal photoreceptor layer in CSC correlated closely with visual acuity loss. Moon et al. [16] demonstrated disintegrity of the junction between foveal outer and inner photoreceptor layer as one of the factors limiting visual improvement with PDT for CSC. In our previous report, we demonstrated that conventional BCVA of patients who underwent hdPDT for chronic CSC was significantly better at 3, 6, and 12 months than at baseline [10]. Moreover, the time course of conventional BCVA improvement was almost the same as that of ellipsoid zone recovery on OCT, suggesting that conventional BCVA was associated with recovery of the ellipsoid zone. In the current study, the time course of ellipsoid zone recovery almost coincided with the time course of LLVA improvement. This finding suggests that recovery of the ellipsoid zone also may play a part in improving LLVA after hdPDT. In addition, the recovery of interdigitation zone was delayed compared with the ellipsoid zone, and the interdigitation zone did not recover in 2 eyes within one year after PDT. In this study, the LLVA and ellipsoid zone improved with a similar course, while the course of interdigitation zone improvement differed from that of LLVA. We speculate that the ellipsoid zone is more strongly related to BCVA and LLVA. Further studies with longer follow-up and larger sample size are needed to clarify the relationship between LLVA and interdigitation zone integrity.

Our study has several limitations. The small sample may have limited the statistical power of our analysis. Another limitation is the lack of a control group not treated with hdPDT. A future study with a larger number of patients and longer follow-up period may validate the efficacy of hdPDT in improving LLVA.

In conclusion, we demonstrated that hdPDT appears to be effective in improving LLVA in patients with chronic CSC. Our results additionally suggest that LLVA may be one of the markers indicating efficacy of hdPDT for CSC cases with relatively good preoperative conventional BCVA.

## Figures and Tables

**Figure 1 jcm-09-03980-f001:**
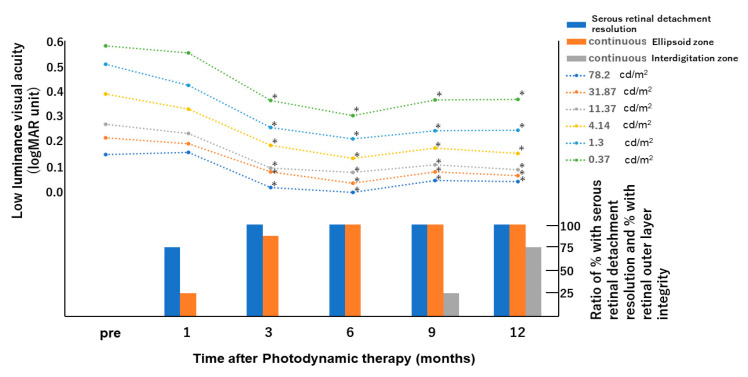
Mean change from baseline of low luminance visual acuity (LLVA), percentage of serous retinal detachment resolution, and integrity of retinal outer layer determined on images of horizontal optical coherence tomographic scans before and after half-dose photodynamic therapy. LLVA is significantly better at 3, 6, and 12 months after surgery than at baseline, at all luminance levels. * *p* < 0.05. The time courses of recovery of the ellipsoid zone and improvement of LLVA are almost the same.

**Figure 2 jcm-09-03980-f002:**
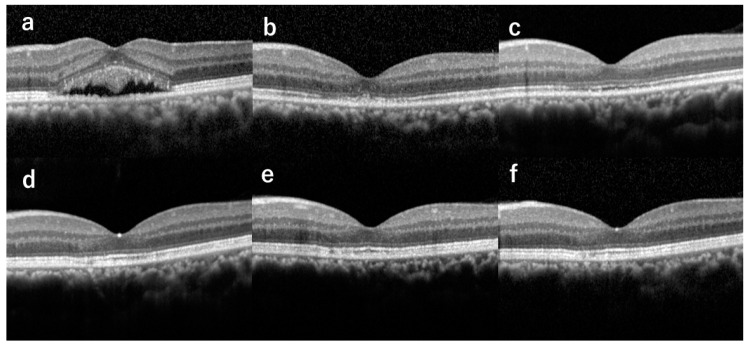
Optical coherence tomography images of the left eye of a 41-year-old male (Case 1) with chronic central serous chorioretinopathy showing serial changes of the ellipsoid zone (EZ) and interdigitation zone (IZ), from before half-dose photodynamic therapy (hdPDT) to complete recovery of the two zones after hdPDT. In this case, the serous retinal detachment was completely absorbed at 1 month after hdPDT. (**a**) Before hdPDT, EZ and IZ at the fovea were both absent. (**b**) At 1 month after hdPDT, EZ and IZ were both absent. (**c**) At 3 months, EZ was continuous and IZ was absent. (**d**) At 6 months, EZ was continuous and IZ was fragmented. (**e**) At 9 months, EZ was continuous and IZ was fragmented. (**f**) At 12 months, EZ and IZ were both continuous.

**Table 1 jcm-09-03980-t001:** Demographics, status of serous retinal detachment, and OCT findings before and after treatment.

Case	Age	Gender	Resolution of SRD					Ellipsoid Zone						Interdigitation Zone					
			1M	3M	6M	9M	12M	Pre	1M	3M	6M	9M	12M	Pre	1M	3M	6M	9M	12M
1	41	male	Yes	Yes	Yes	Yes	Yes	No	No	Yes	Yes	Yes	Yes	No	No	No	No	No	Yes
2	33	male	Yes	Yes	Yes	Yes	Yes	No	No	No	Yes	Yes	Yes	No	No	No	No	No	Yes
3	38	male	Yes	Yes	Yes	Yes	Yes	No	No	Yes	Yes	Yes	Yes	No	No	No	No	No	Yes
4	44	male	No	Yes	Yes	Yes	Yes	No	No	Yes	Yes	Yes	Yes	No	No	No	No	No	Yes
5	45	male	Yes	Yes	Yes	Yes	Yes	No	No	Yes	Yes	Yes	Yes	No	No	No	No	No	No
6	51	male	Yes	Yes	Yes	Yes	Yes	No	Yes	Yes	Yes	Yes	Yes	No	No	No	No	No	No
7	56	male	No	Yes	Yes	Yes	Yes	No	No	Yes	Yes	Yes	Yes	No	No	No	No	Yes	Yes
8	42	male	Yes	Yes	Yes	Yes	Yes	No	Yes	Yes	Yes	Yes	Yes	No	No	No	No	Yes	Yes

M: months; SRD: serous retinal detachment; OCT: optical coherence tomography.

**Table 2 jcm-09-03980-t002:** Conventional best corrected visual acuity and low luminance visual acuity at each follow-up visit.

		Pre	1M	3M	6M	9M	12M
Conventional BCVA ^a^		−0.05 ± 0.10	−0.03 ± 0.05	−0.06 ± 0.05	−0.06 ± 0.05	−0.08 ± 0.04	−0.07 ± 0.05
Low luminance visual acuity	78.20 cd/m^2^	0.14 ± 0.12	0.15 ± 0.08	0.01 ± 0.09	−0.006 ± 0.10	0.04 ± 0.08	0.04 ± 0.07
	31.87 cd/m^2^	0.21 ± 0.12	0.18 ± 0.08	0.08 ± 0.06	0.03 ± 0.08	0.07 ± 0.10	0.06 ± 0.05
	11.37 cd/m^2^	0.27 ± 0.11	0.23 ± 0.10	0.09 ± 0.07	0.07 ± 0.10	0.10 ± 0.07	0.08 ± 0.06
	4.14 cd/m^2^	0.38 ± 0.13	0.32 ± 0.09	0.18 ± 0.06	0.13 ± 0.09	0.17 ± 0.10	0.15 ± 0.09
	1.30 cd/m^2^	0.51 ± 0.12	0.42 ± 0.10	0.25 ± 0.07	0.21 ± 0.09	0.24 ± 0.10	0.24 ± 0.09
	0.37 cd/m^2^	0.58 ± 0.08	0.55 ± 0.13	0.36 ± 0.07	0.30 ± 0.09	0.36 ± 0.11	0.36 ± 0.09

M: months; BCVA: best corrected visual acuity. Data are expressed as mean ± standard deviation. ^a^ Conventional best corrected visual acuity measurement was performed under photopic condition with background luminance of 200 cd/ m^2^ (standardized chart luminance is 80 to 320 cd/m^2^).

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
