# Peer review of "Improvement of Low Luminance Visual Acuity in Patients with Chronic Central Serous Chorioretinopathy after Half-Dose Verteporfin Photodynamic Therapy"

_jcm, 2020, doi:10.3390/jcm9123980_

Round 1

Reviewer 1 Report

Thank you for allowing me to review the manuscript entitled “Improvement of Low Luminance Visual Acuity in Patients with Chronic Central Serous Chorioretinopathy after Half-dose Verteporfin Photodynamic Therapy”. This paper focuses on the time courese of low luminance visual acuity (LLVA) in eyes with chronic central serous chorioretinopathy after half-dose verteporfin photodynamic therapy. Below I discuss some issues, which hopefully can help you improve the study.

Comments:

  1. Please define the restoration of the photoreceptor layer in more detail. For example, 300um around the fovea.
  2. In what order did authors measure visual acuities? From low luminance? Please explain.
  3. In figure 1, did authors mean percentage of ‘eyes’ with serous retinal detachment resolution and integrity of retinal outer layer?
  4. Please discuss about the relationship between interdigitation zone and low luminance visual acuities.

Author Response

Reviewer 1

Thank you for allowing me to review the manuscript entitled “Improvement of Low Luminance Visual Acuity in Patients with Chronic Central Serous Chorioretinopathy after Half-dose Verteporfin Photodynamic Therapy”. This paper focuses on the time courese of low luminance visual acuity (LLVA) in eyes with chronic central serous chorioretinopathy after half-dose verteporfin photodynamic therapy. Below I discuss some issues, which hopefully can help you improve the study.

 Comments:

  • Please define the restoration of the photoreceptor layer in more detail. For example, 300um around the fovea.

Response:

We would like to thank the reviewer for the comments.

Following the reviewer’s comment, we have added more detail to the definition of restoration of the photoreceptor layer as follows,

Lines 79-80

Restoration of the photoreceptor layer was defined as reconstruction of the continuous back-reflection lines corresponding to the ellipsoid zone and the interdigitation zone at the central fovea.

  • In what order did authors measure visual acuities? From low luminance? Please explain.

Response:

We thank the reviewer for the comment.

Low luminance visual acuity (LLVA) was measured from the lowest background to higher background sequentially.

We have added the explanation in the Methods as follows.

Lines 75-76

LLVA was measured starting from the lowest background and increasing sequentially to higher background luminance.

  • In figure1, did authors mean percentage of ‘eyes’ with serous retinal detachment resolution and integrity of retinal outer layer?

Response:

We thank the reviewer for the comment.

As the reviewer correctly pointed out, the bar graphs show not only “percent of eyes with retinal outer layer integrity” but also “percent of eyes with serous retinal detachment resolution”. We apologize for the omission.

We have corrected the label in Figure 1.

  • Please discuss about the relationship between interdigitation zone and low luminance visual acuities.

Response:

We thank the reviewer for pointing out this issue.

To date, interdigitation zone integrity has been shown to correlate with BCVA in various disorders. The detailed mechanism is still unclear.

In this study, although LLVA and ellipsoid zone improved with a similar course, the course of interdigitation zone improvement differed from that of LLVA. We speculate that the ellipsoid zone is more strongly related to BCVA and LLVA. Further studies with longer follow-up and larger sample size are needed to clarify the relationship between LLVA and interdigitation zone integrity.

We have added the comments as follows in Discussion.

Lines 157-163

In addition, the recovery of interdigitation zone was delayed compared with the ellipsoid zone, and the interdigitation zone did not recover in 2 eyes within one year after PDT. In this study, the LLVA and ellipsoid zone improved with a similar course, while the course of interdigitation zone improvement differed from that of LLVA. We speculate that the ellipsoid zone is more strongly related to BCVA and LLVA. Further studies with longer follow-up and large sample size are needed to clarify the relationship between LLVA and interdigitation zone integrity.

Reviewer 2 Report

Line 39: Is serous retinal detachment (SRD) at the macula the only characteristic feature of CSC?

Line 52: Metamorphopsia cannot improve as this is an unfavorable symptom.

The equipment used for ICGA FA should be specified.

Who assessed the ellipsoid zone and what method was used?

Lines 78-79: Restoration of the photoreceptor layer was defined as reconstruction of the continuous back reflection lines corresponding to the ellipsoid zone and the interdigitation zone at the fovea - czy było oceniane ilściowo/ jakościowo??

Did the authors you perform quantitative and qualitative analysis?

I strongly suggest adding the “before” and “after” images for SD OCT, depicting the restoration of the photoreceptor layer.

The lack of a control group that would include patients who were not treated with half-dose verteporfin photodynamic therapy is a considerable limitation of the study.

What was the exact disease duration before inclusion in the study? The authors only provide information that CSC lasted longer than 6 months. Did the authors assess how the duration of CSC correlated with improved complete blood count parameters and function?

Although VA is preserved relatively well in spite of SRD in the macula, patients often complain of various visual disturbances such as metamorphopsia, relative central scotoma and micropsia [5] what significantly affects their quality of life.

Consider citing this study referring to the quality of patients with CSC: Karska-Basta I, Pociej-Marciak W, Chrząszcz M, Żuber-Łaskawiec K, Sanak M, Romanowska-Dixon B. Quality of life of patients with central serous chorioretinopathy - a major cause of vision threat among middle-aged individuals. Archives of Medical Science. 2020: In press. Doi: https://doi.org/10.5114/aoms.2020.92694

Language

The paper is well structured, and the scientific terminology is appropriate and consistent. The title is sufficiently informative and concise. The purpose of the research is clearly stated. The methodology is described in sufficient detail to be reproduced by another investigator. The level of statistical significance has been defined, and ethical issues have been addressed. The results are provided in an appropriate section, and there are no redundancies and repetitions in the Discussion section. The conclusions are clear. The level of the English language is good, although minor errors and inconsistencies could be improved. The examples are listed below:

Use of articles:

Line 36: “with increase in logarithm of illumination” should read “with an increase in the logarithm of illumination”

Line 48: “patients had visual difficulties in environment with lower luminance” should read “patients had visual difficulties in an environment with lower luminance”

Tenses:

Line 57: “Inclusion criteria include the presence of” should read “Inclusion criteria included the presence of” (simple past tense is preferred for the reporting of methods/results).

Line 134: “To date, there is no study that evaluates LLVA after hdPDT” should read “To date, no studies have evaluated LLVA after hdPDT” or “To date, there have been no studies evaluating LLVA after hdPDT”.

Word choice:

Line 19:  “were evaluated before, and at 1, 3, 6, 9 and 12 months” should read “were evaluated at baseline and at 1, 3, 6, 9 and 12 months” (“baseline” is commonly used to mean “before treatment”).

Redundancy:

Line 94: “The conventional BCVA improved significantly from -0.06 ± 0.10 logMAR at baseline to -0.16 ± 0.03 logMAR at 1 year (P = 0.019)” should read “The conventional BCVA improved from -0.06 ± 0.10 logMAR at baseline to -0.16 ± 0.03 logMAR at 1 year (P = 0.019)” (“significantly” is redundant if the exact P value is provided and the significance level has been defined in the Methods section).

Punctuation:

Line 144: “Several studies have shown that the status of the ellipsoid zone is associated with VA. in CSC patients [9.14.15]. Piccolino et al. [14] showed…” should read “Several studies have shown that the status of the ellipsoid zone is associated with VA in CSC patients [9,14,15]. Piccolino et al. [14] showed …” (please note the inappropriate use of a full stop after VA and for the set of references in brackets).

Author Response

Reviewer 2

Comments and Suggestions for Authors

Line 39: Is serous retinal detachment (SRD) at the macula the only characteristic feature of CSC?

Response:

We would like to thank the reviewer for the comments.

As the reviewer pointed out, other than serous retinal detachment, central serous chorioretinopathy is characterized also by retinal pigment epithelial detachment.

We have changed the explanation in Introduction as follows.

Lines 39-40

Central serous chorioretinopathy (CSC) is characterized by serous retinal detachment (SRD) and retinal pigment epithelial detachment at the macula.

Line 52: Metamorphopsia cannot improve as this is an unfavorable symptom.

Response:

We thank the reviewer for the comment.

In this sentence (lines 50-53), we cited several reports on the effectiveness of half-dose verteporfin photodynamic therapy (hdPDT) for CSC, including our previous report that metamorphopsia improved significantly after half-dose photodynamic therapy (10. Fujita K, et al. Retina, 2014). However, we agree with the reviewer that metamorphopsia may not improve completely in some patients.

The equipment used for ICGA FA should be specified.

Response:

We thank the reviewer for the comment.

We have added the manufacturer of ICGA in Methods.

Lines 60-61

…….indocyanine green angiography (ICGA) (TRC-50IX; Topcon Corp., Tokyo, Japan)……

Who assessed the ellipsoid zone and what method was used?

Response:

We thank the reviewer for the comment.

Qualitative OCT findings were evaluated independently by two retina specialists (KF and KS). When there were discrepancies between the two retina specialists, a third specialist (YI) made the final decision.

We have added the assessment the OCT findings in Methods.

Lines 81-83

Qualitative OCT findings were evaluated independently by two retina specialists (KF and KS). When there were discrepancies between the two retina specialists, a third specialist (YI) made the final decision.

Lines 78-79: Restoration of the photoreceptor layer was defined as reconstruction of the continuous back reflection lines corresponding to the ellipsoid zone and the interdigitation zone at the fovea – czy było oceniane ilściowo/ jakościowo??

Response:

We are sorry not to be able to response to this comment, because we do not understand the language.

Did the authors you perform quantitative and qualitative analysis?

Response:

We thank the reviewer for the comment.

For assessment of restoration of the photoreceptor layer (reconstruction of the continuous back reflection lines corresponding to the ellipsoid zone and the interdigitation zone at the central fovea), we performed qualitative analysis on SD-OCT images. We have given more details of the evaluation, as described in our response to a comment above.

Lines 81-83

Qualitative OCT findings were evaluated independently by two retina specialists (KF and KS). When there were discrepancies between the two retina specialists, a third specialist (YI) made the final decision.

I strongly suggest adding the “before” and “after” images for SD OCT, depicting the restoration of the photoreceptor layer.

Response:

We appreciate the reviewer’s comment.

As the authors have moved from the hospital where this study was performed to other hospitals, we are not able to obtain images of those cases in this study. We apologize for not able to add the images.

The lack of a control group that would include patients who were not treated with half-dose verteporfin photodynamic therapy is a considerable limitation of the study.

Response:

We thank the reviewer for pointing out this issue. Following the reviewer’s comment, we have added the limitation this issue in Discussion.

Lines 165-166

The small sample may have limited the statistical power of our analysis. Another limitation is the lack of a control group not treated with hdPDT.

What was the exact disease duration before inclusion in the study? The authors only provide information that CSC lasted longer than 6 months. Did the authors assess how the duration of CSC correlated with improved complete blood count parameters and function?

Response:

We thank the reviewer for pointing out this issue.

Most patients with CSC do not notice their symptoms even if they have retinal detachment. It is difficult to identify the duration correctly.

Because we did not perform blood tests in this study, the correlation between the duration of CSC and complete blood count was unknown.

Although VA is preserved relatively well in spite of SRD in the macula, patients often complain of various visual disturbances such as metamorphopsia, relative central scotoma and micropsia [5] what significantly affects their quality of life.

Consider citing this study referring to the quality of patients with CSC: Karska-Basta I, Pociej-Marciak W, Chrząszcz M, Żuber-Łaskawiec K, Sanak M, Romanowska-Dixon B. Quality of life of patients with central serous chorioretinopathy - a major cause of vision threat among middle-aged individuals. Archives of Medical Science. 2020: In press. Doi: https://doi.org/10.5114/aoms.2020.92694

Response:

We thank the reviewer for suggestion.

Indeed, the various visual disturbances of patients with CSC may impact their quality of life. We have cited the suggested reference as follows:

Lines 41-43

Although VA is preserved relatively well in spite of SRD in the macula, patients often complain of various visual disturbances such as metamorphopsia, relative central scotoma and micropsia [5], which may impact patients’ quality of life [6].

Language

The paper is well structured, and the scientific terminology is appropriate and consistent. The title is sufficiently informative and concise. The purpose of the research is clearly stated. The methodology is described in sufficient detail to be reproduced by another investigator. The level of statistical significance has been defined, and ethical issues have been addressed. The results are provided in an appropriate section, and there are no redundancies and repetitions in the Discussion section. The conclusions are clear. The level of the English language is good, although minor errors and inconsistencies could be improved. The examples are listed below:

Use of articles:

Line 36: “with increase in logarithm of illumination” should read “with an increase in the logarithm of illumination”

Response:

We thank the reviewer for the comment.

We have added the articles.

Lines 36-37

…. VA increases rapidly with an increase in the logarithm of illumination until plateau is reached, and this is cone-dominant function.

Line 48: “patients had visual difficulties in environment with lower luminance” should read “patients had visual difficulties in an environment with lower luminance”

Response:

We thank the reviewer for the comment.

We have added the article.

Lines 48-49

These results indicated that CSC patients had visual difficulties in an environment with lower luminance.

Tenses:

Line 57: “Inclusion criteria include the presence of” should read “Inclusion criteria included the presence of” (simple past tense is preferred for the reporting of methods/results).

Response:

We thank the reviewer for the comment.

We have corrected the tense.

Line 57

Inclusion criteria included the presence of SRD involving the fovea depicted on spectral-domain optical coherence tomography (OCT) (SD-OCT; Heidelberg Engineering, Heidelberg, Germany) and abnormally dilated choroidal vasculature with hyperpermeability observed on indocyanine green angiography (ICGA) before hdPDT.

Line 134: “To date, there is no study that evaluates LLVA after hdPDT” should read “To date, no studies have evaluated LLVA after hdPDT” or “To date, there have been no studies evaluating LLVA after hdPDT”.

Response:

We thank the reviewer for the comment.

We have revised the sentence.

Line 137

To date, no studies have evaluated LLVA after hdPDT.

Word choice:

Line 19 “were evaluated before, and at 1, 3, 6, 9 and 12 months” should read “were evaluated at baseline and at 1, 3, 6, 9 and 12 months” (“baseline” is commonly used to mean “before treatment”).

Response:

We thank the reviewer for the comment.

We have changed the term.

Line 19

The best-corrected visual acuity, LLVA and optical coherence tomography (OCT) findings were evaluated at baseline and at 1, 3, 6, 9 and 12 months after hdPDT.

Redundancy:

Line 94: “The conventional BCVA improved significantly from -0.06 ± 0.10 logMAR at baseline to -0.16 ± 0.03 logMAR at 1 year (P = 0.019)” should read “The conventional BCVA improved from -0.06 ± 0.10 logMAR at baseline to -0.16 ± 0.03 logMAR at 1 year (P = 0.019)” (“significantly” is redundant if the exact P value is provided and the significance level has been defined in the Methods section).

Response:

We thank the reviewer for the comment.

We have omitted the word significant.

Line 98

The conventional BCVA improved from -0.06 ± 0.10 logMAR at baseline to -0.16 ± 0.03 logMAR at 1 year (P = 0.019)

Punctuation:

Line 144: “Several studies have shown that the status of the ellipsoid zone is associated with VA. in CSC patients [9.14.15]. Piccolino et al. [14] showed…” should read “Several studies have shown that the status of the ellipsoid zone is associated with VA in CSC patients [9,14,15]. Piccolino et al. [14] showed …” (please note the inappropriate use of a full stop after VA and for the set of references in brackets).

Response:

We thank the reviewer for comment.

We have corrected the punctuations.

Lines 147-148

Several studies have shown that the status of the ellipsoid zone is associated with VA in CSC patients [9,14,15]. Piccolino et al. [14] showed that impairment of the foveal photoreceptor layer in CSC correlated closely with visual acuity loss.

Reviewer 3 Report

The authors present visual function (LLVA) on 8 patients with central serous chorioretinopathy (CSC) after half-dose verteporfin photodynamic therapy (hdPDT) over 1, 3, 6, 9 and 12 months after hdPDT.

This study is just too small sample size to assess any benefit on LLVA despite it being measured at six levels of background luminance.

The test to test variability n CSCR is large especially in such chronic cases. 

Author Response

Reviewer 3

Comments and Suggestions for Authors

The authors present visual function (LLVA) on 8 patients with central serous chorioretinopathy (CSC) after half-dose verteporfin photodynamic therapy (hdPDT) over 1, 3, 6, 9 and 12 months after hdPDT.

This study is just too small sample size to assess any benefit on LLVA despite it being measured at six levels of background luminance.

The test to test variability n CSCR is large especially in such chronic cases.

Response: We would like to thank the reviewer for the comments.

As the reviewer pointed out, this study had a small sample size. We considered this study as a pilot study. Although the results were preliminary, we were able to evaluate the results of hdPDT for chronic CSC not only regarding conventional morphological changes and visual acuity, but also low luminance visual acuity and recovery of integrity of retinal outer layer. The results would be validated in a large-scale study with long-term follow-up.